# Early Childhood Temperamental Trajectories following Very Preterm Birth and Their Association with Parenting Style

**DOI:** 10.3390/children9040508

**Published:** 2022-04-03

**Authors:** Irene Lovato, Lucy D. Vanes, Chiara Sacchi, Alessandra Simonelli, Laila Hadaya, Dana Kanel, Shona Falconer, Serena Counsell, Maggie Redshaw, Nigel Kennea, Anthony David Edwards, Chiara Nosarti

**Affiliations:** 1Centre for the Developing Brain, School of Biomedical Engineering & Imaging Sciences, King’s College London, London SE1 7EH, UK; irene.lovato@studenti.unipd.it (I.L.); lucy.vanes@kcl.ac.uk (L.D.V.); laila.hadaya@kcl.ac.uk (L.H.); dana.kanel@kcl.ac.uk (D.K.); shona.falconer@kcl.ac.uk (S.F.); serena.counsell@kcl.ac.uk (S.C.); ad.edwards@kcl.ac.uk (A.D.E.); 2Department of Developmental Psychology and Socialization, University of Padova, 35151 Padova, Italy; chiara.sacchi@unipd.it (C.S.); alessandra.simonelli@unipd.it (A.S.); 3Department of Neuroimaging, Institute of Psychiatry, Psychology and Neuroscience, King’s College London, London SE5 8AF, UK; 4Department of Child and Adolescent Psychiatry, Institute of Psychiatry, Psychology and Neuroscience, King’s College London, London SE5 8AF, UK; 5National Perinatal Epidemiology Unit, University of Oxford, Oxford OX3 7LF, UK; maggie.redshaw@outlook.com; 6Neonatal Unit, St George’s Hospital, London SW17 0QT, UK; nigel.kennea@stgeorges.nhs.uk

**Keywords:** very preterm birth, temperament, dysfunctional parenting

## Abstract

Childhood temperament is an early characteristic shaping later life adjustment. However, little is currently known about the stability of early temperament and its susceptibility to the environment in children born very preterm (VPT; <33 weeks’ gestation). Here, we investigated infant-to-childhood temperamental trajectories, and their interaction with parental practices, in VPT children. Maternal reports of infant temperament were collected in 190 infants (mean age: 11.27 months; range 9–18 months) enrolled in the longitudinal Evaluation of Preterm Imaging (ePrime; Eudra: CT 2009-011602-42) study, using the ePrime questionnaire on infant temperament. At 4–7 years of age, further assessments of child temperament (Children’s Behavior Questionnaire—Very Short Form) and parenting style (Arnold’s Parenting Scale) were conducted. Results showed that more difficult temperament in infancy was associated with increased Negative Affectivity in childhood, regardless of parenting practices. This lends support to the stability of early temperamental traits reflecting negative emotionality. In contrast, a lax parenting style moderated the relationship between easy infant temperament and Negative Affectivity at 4–7 years, such that an easier infant temperament was increasingly associated with higher childhood Negative Affectivity scores as parental laxness increased. These results highlight a potential vulnerability of VPT infants considered by their mothers to be easy to handle, as they may be more susceptible to the effects of suboptimal parenting in childhood.

## 1. Introduction

Very preterm birth (<33 weeks’ gestation) represents approximately 15% of all preterm births (<37 weeks’ gestation), which are currently estimated by the World Health Organization to constitute between 5% and 18% of all births [1,2]. Therefore, attention has increasingly focused on the quality of life of survivors, who are at high risk of developing a range of adverse short- and long-term sequelae [3]. The neurodevelopmental outcomes associated with preterm birth have been summarised into a “preterm behavioural phenotype” characterised by an increased risk of exhibiting inattention, anxiety and socio-emotional difficulties [4].

The identification of specific vulnerabilities associated with preterm birth have stimulated a growing interest in finding early markers that could be used to predict neurodevelopmental outcomes. Temperament is an important aspect of early development, shaping indicators of later life adjustment, including mental health and school adaptation (for a review, see [5]). In a generally accepted summary of several conceptualisations of temperament [6], this refers to “relatively consistent, basic dispositions inherent in the person that underlie and modulate the expression of activity, reactivity, emotionality and sociability”. These biologically based dispositions appear in infancy, and are considered to be relatively stable over time, both when assessed via maternal reports or laboratory measures [7,8,9,10,11]. In preterm individuals, temperamental traits in middle childhood have been shown to be predictive of later behavioural outcomes [12]. However, only a few studies to date have investigated whether temperament in infancy is a stable predictor of childhood temperament in those born preterm, and as such may constitute a useful early indicator of behavioural risk. With an increasing interest in early detection and intervention in developmental psychopathology [13,14], understanding the development of temperament from infancy to childhood following preterm birth may shed light on early mechanisms shaping later outcomes in this population.

Despite its relative stability, children’s temperament is also known to interact in a complex way with environmental and parental factors, with children shaping their development through cascades of individual–context bidirectional effects [15,16,17]. In particular, childhood temperament has been associated with parents’ health and parenting style. For example, negative emotionality has been linked to a strict and controlling maternal discipline, and to an authoritarian parenting style [18,19,20,21]. Similarly, high levels of activity have been associated with maternal strictness [18], but also with a parenting style characterised by authority, warmth and clear rules [21], possibly reflecting parental acceptance of higher activity levels in their children. Finally, children’s inhibition and shyness have been related to maternal overprotectiveness [22,23], although the direction of the relationship remains to be ascertained. Mothers may tend to display overprotective responses to children’s shyness, fear and inhibition [24], or, alternatively, overprotective mothers may be less encouraging of their children’s independence, which in turn may exacerbate feelings of social anxiety [22]. It has been suggested that children’s inhibited reactions and mothers’ overprotective behaviours may reinforce one another over time in an “anxious-coercive cycle” that could lead to the development of anxiety problems [25]. Nevertheless, difficult temperamental characteristics have also been associated with a positive parental style, with parents making positive efforts, as well as providing more contact and stimulation, with their irritable and demanding children [26]. Furthermore, specific temperamental traits could elicit different parenting styles in different parents, and parenting might influence a child’s temperament both via effects on early brain development or via a mechanism of parental modelling of emotional and social behaviours [27].

Child temperament and parenting are thought to interact in shaping child development [23,28,29,30,31,32]. In general, children with a more difficult temperament are more vulnerable to the effects of poor-quality parenting, but also benefit more from positive parenting, in line with the theory of differential susceptibility [33]. These differences in susceptibility can manifest in several aspects of negative childhood adjustment, including externalising and internalising problems, as well as poorer social and cognitive competence [34].

The literature suggests that temperament may differ between preterm (predominantly very preterm and extremely preterm; <28 weeks’ gestation) children and term-born controls, especially regarding the level of activity and self-regulation (for a review, see [35]). In particular, preterm children show a higher activity level, as well as lower attention span and focus, compared with their term-born peers [35]. Both these temperamental traits may be early indicators of behavioural difficulties [12], underlying preterm children’s higher risk of inattention and externalising behaviour problems [36,37]. Preterm children’s temperament development may be affected by maternal prenatal stress, which has been associated with alterations in hormones and neurotransmitters in utero [38], influencing infants’ brain development [39], as well as early emotional responses and reactions to stimuli [40]. We have in fact previously found associations between structural and functional properties of limbic brain regions in the neonatal period and the capacity to regulate emotions later in childhood in very preterm individuals [41,42]. Furthermore, interactions between preterm and full-term child–parent dyads are thought to differ, as very and extremely preterm children show poorer emotional and behavioural regulation, and their parents exhibit more difficulties in being responsive and sensitive to their children’s needs. [43,44]. Understanding temperament-by-parenting interactions on behaviour in preterm samples has therefore been of particular interest. Clark and colleagues [43] observed that extremely preterm children aged 2–4 years had poorer self-regulation compared to term-born controls and that, in the extremely preterm group, a less sensitive parenting style was one of the strongest predictors of self-regulation impairment. Poehlmann et al. [45] found that preterm low-birth-weight infants (<2500 g at birth) were more prone to distress and exhibited high levels of externalising behaviours at 24 months only if they experienced angry and critical parenting at 9 months. Proneness to distress in infancy was also found to be a risk factor for internalising problems at 24 months, but only when accompanied by an intrusive and anxious parenting style. Furthermore, more difficult infants displayed lower Effortful Control at 24 months if they experienced a parenting style characterised by anxiety and intrusiveness [45]. Lastly, Whiteside-Mansell and colleagues [46] reported that preterm low-birth-weight infants with difficult temperament were at increased risk of developing externalising problems when exposed to family conflicts.

To date, there is little research using a longitudinal design to examine whether very early markers of temperament are predictive of later temperamental outcomes following very preterm birth, and to study how these trajectories may be influenced by parenting style. In this study, we therefore investigated whether maternal perceptions of infant temperament were associated with childhood temperament assessed between 4 and 7 years of age, in a sample of very preterm children. We used a simple conceptualisation of infant temperament, defined as being either “easy or “difficult” [11], where an easy infant is responsive and content, while a difficult infant is demanding and tough to deal with. Childhood temperament was captured in terms of standardised measures of Negative Affectivity (overall negative outlook of oneself and the surrounding world), Surgency (a disposition towards impulsivity, intense pleasure seeking and high activity levels) and Effortful Control (a child’s capacity to focus and shift attention, intentionally inhibit a response and respond to low-intensity stimulation and reward) [47]. Furthermore, we explored whether suboptimal parenting influenced temperamental trajectories from infancy to childhood. We hypothesised that (a) early maternal perceptions of infant temperament would be predictive of temperamental traits during childhood, as temperament itself is defined as relatively stable, and (b) infants with a difficult temperament would be more vulnerable to the effects of suboptimal parenting than infants with an easy temperament.

## 2. Materials and Methods

### 2.1. Design

A longitudinal study was conducted to evaluate stability of maternal perception of temperamental dimensions between infancy and childhood, and the influence of suboptimal parenting style on these trajectories. Mothers’ reports of child’s temperament were collected when children were 9–18 months and 4–7 years, while parenting style was assessed with maternal self-report when children were 4–7 years.

### 2.2. Sample

Study participants were 511 very preterm-born infants recruited at birth into the Evaluation of Preterm Imaging study (ePrime, Eudra: CT 2009-011602-42) between April 2010 and July 2013 from hospitals in the North and South-West London Perinatal Network [48]. Eligibility criteria were birth before 33 weeks’ gestation, mothers being over 16 years and not hospital inpatients. Exclusion criteria included major congenital malformation and parents being unable to speak English or subject to child protection proceedings. Of the ePrime cohort, a convenience sample of 251 children (corresponding to 82% of 306 who were eligible and invited to participate) underwent a neurodevelopmental follow-up assessment between the ages of 4 and 7 years (67% age 4, 17% age 5, 13% age 6 and 3% age 7). Written informed consent was given by the children’s carer(s), following protocols approved by Stanmore Research Ethics Committee (14/LO/0677). This study was conducted in line with the Code of Ethics of the World Medical Association (Declaration of Helsinki). 

Here, we report results from 190 participants with complete temperamental data in infancy and at 4–7 years follow-up. Data for both child temperament and parenting style were available for 101 subjects.

### 2.3. Socio-Demographic Data

An Index of Multiple Deprivation (IMD) score was derived from parental postcode at the time of infant birth (Department for Communities and Local Government 2011; https://tools.npeu.ox.ac.uk/imd/ (accessed on 3 March 2015)). The IMD measures social risk by comparing seven domains of deprivation within each neighbourhood: income, employment, education skills and training, health and disability, barriers to housing and services, living environment and crime. A higher IMD value indicates higher deprivation.

### 2.4. Assessment of Infant Temperament 

Child’s temperament was assessed in infancy (mean age = 11.27 months, range = 9–18.4 months) via a qualitative maternal questionnaire (ePrime questionnaire on infant temperament). The questionnaire is based on an adjective checklist originally designed in order to assess parents’ perception of their infant while in the neonatal unit [49] and in the months thereafter [50,51], consisting of both positive and negative terms. 

The ePrime questionnaire on infant temperament requires the parent to circle as many adjectives as possible from a list that they believe best describe their baby at time of assessment. While this type of checklist has been previously used to assess the proportion of positive and negative adjectives parents use to describe their infant [49,50,51], here we used it to derive scores for “easy” and “difficult” temperament by evaluating only adjectives which incorporate aspects of the infant’s activity level or behaviour (such as “settled” or “agitated”), but not purely descriptive adjectives (such as “beautiful” or “small”). Thus, the “easy” temperament category includes adjectives such as settled, cuddly, active and responsive, while the “difficult” temperament category represents infants that are demanding or tough to deal with (for example being agitated or, alternatively, unresponsive and inactive).

A full description of the ePrime questionnaire on infant temperament can be found in Appendix A.

### 2.5. Follow-Up Assessment at 4–7 Years

#### 2.5.1. Children’s Behavior Questionnaire—Very Short Form

The Children’s Behavior Questionnaire—Very Short Form (CBQ-VSF) is a 36-item parent-rated questionnaire assessing temperament, which is validated for children aged 3–8 [47]. The CBQ is a widely used tool in preterm research [35], and it is based on Rothbart’s well-accepted psychobiological theory of temperament, which defined temperament as relatively stable and biologically based individual differences in levels of reactivity and self-regulation [52]. In this questionnaire, parents are asked to describe their children in situations occurring in everyday life using a 7-point scale from 1 (extremely untrue of your child) to 7 (extremely true of your child); higher scores therefore indicate higher levels of a temperamental trait. The CBQ-VSF provides scores for three temperamental traits: Surgency, Negative Affectivity and Effortful Control. Surgency is characterised by a disposition toward positive emotions, high activity level and a rapid approach to potential rewards [53]. Negative Affectivity is characterised by shyness, discomfort, anger–frustration, fear, sadness and un-soothability [9]. Effortful Control encompasses voluntary attentional focusing, attentional shifting, inhibitory and activational control of behaviours [52]. All CBQ-VSF subscales have been shown to exhibit adequate internal consistency (Cronbach’s alpha > 0.66) [47].

#### 2.5.2. Parenting Scale

The Parenting Scale is a 30-item rating scale that measures dysfunctional parenting in discipline situations [54]. Parents are asked to indicate their tendency to use specific discipline practices using a 7-point scale, with dysfunctional parenting style at one end, and supporting parenting style at the other. An example is “When my child misbehaves, I raise my voice or yell; or I speak to my child calmly”. Three different suboptimal parenting styles are evaluated, as well a global index of dysfunctional parenting. Higher scores indicate a poorer disciplinary parenting style. Overreactivity indicates authoritarian and coercive discipline practices. Laxness, in contrast, describes a permissive parent who is inconsistent in providing discipline. Finally, Verbosity refers to a parenting style characterised by lengthy and ineffective verbal reprimands. All subscales of the Parenting Scale have been shown to exhibit adequate internal consistency (Cronbach’s alpha > 0.63) [54].

### 2.6. Statistical Analysis

Statistical analyses were performed using R [55] and RStudio [56]. Throughout, we chose a multivariate regression approach (whereby childhood temperament variables were initially included simultaneously as dependent variables, to be predicted by infant temperament, parenting and interactions thereof). This analysis tests the null hypothesis that the coefficient for each predictor is equal to zero across all dependent variables. This method allowed for a principled approach for choosing suitable follow-up analyses, thus reducing the overall number of tests in the context of a large number of possible variable combinations and thereby accounting for likely error inflation due to multiple testing [57,58].

Firstly, a multivariate regression analysis was run in order to explore the relationship between infant and childhood maternal perception of temperament. The three subscales of the CBQ were simultaneously included as dependent variables, and the two infant temperament variables (easy and difficult) were included as predictors in the same model. Non-collinearity of these variables was evaluated using correlation analyses; indeed, there was no significant relationship between easy and difficult infant temperament (*r* = −0.067, *df* = 188, *p* = 0.356). The analysis was performed including the following covariates: sex, gestational age (GA) at birth, IMD score and age at follow-up assessment. Following a significant effect of either easy or difficult temperament (at a significance threshold of *p* = 0.05), individual follow-up univariate regression models for each CBQ subscale (Negative Affectivity, Surgency and Effortful Control) were conducted to elucidate the nature and direction of the association. These follow-up tests were corrected for multiple comparisons using Bonferroni correction for three separate models (*p* = 0.05/3 = 0.017).

To test whether adding parenting style to the original model predicting temperamental outcomes at 4–7 years significantly improved model prediction, multivariate model comparisons for each Parenting Scale subscale (Laxness, Overreactivity and Verbosity) were performed. For each parenting subscale, a multivariate null model (M0), predicting childhood temperament from easy and difficult infant temperament as described above (controlling for sex, GA at birth, IMD score and age at follow-up), was separately tested against two more complex models (M1 and M2) using likelihood ratio F-tests. M1 additionally included the respective parenting subscale and its interaction with easy temperament, while M2 included the parenting subscale and its interaction with difficult temperament. The significance threshold was adjusted using Bonferroni correction for 6 separate model comparisons (one for each parenting variable and separately for its interaction with easy or difficult temperament), resulting in a threshold of *p* = 0.05/6 = 0.008.

Where either M1 or M2 was shown to explain significantly more variance compared to M0, follow-up univariate tests for each CBQ variable were performed. These tests were further Bonferroni-corrected using a significance threshold of 0.008/3 = 0.003. In the case of a significant interaction between continuous variables, these were further elucidated using simple slope analyses (evaluated at ±1 SD and ±1.5 SD from the mean).

Analysis code is available at https://github.com/lucyvanes/preterm-temperament. 

## 3. Results

### 3.1. Sample Characteristics

Socio-demographic characteristics of both samples with complete temperamental data and subsamples with additional parenting data are shown in Table 1. Subsample participants with parenting data (*n* = 101) did not differ significantly from participants with temperamental but without parenting data (*n* = 89) on any key maternal or child characteristics listed in Table 1 (as ascertained via t-tests for continuous and chi-squared tests for categorical variables). One exception was that participants excluded from the parenting analysis (i.e., without parenting data) at 4–7 years follow-up assessment were slightly older than those included (mean = 4.5 vs. 5.4, *t*(188) = 10.53, *p* < 0.001).

### 3.2. Association between Infant and Childhood Temperament

A multivariate regression analysis including the CBQ subscales (Negative Affectivity, Surgency and Effortful Control) as dependent variables, and easy and difficult temperaments as predictors (controlling for sex, GA at birth, IMD score and age at follow-up assessment) showed a significant relationship between a difficult infant temperament and childhood temperament (*F*(3,181) = 4.5, *p* = 0.004), and between an easy infant temperament and childhood temperament (*F*(3,181) = 3.1, *p* = 0.027) (see Table 2 for full multivariate model output).

Follow-up individual linear regressions for each CBQ subscale (see Table 3; Bonferroni-corrected for three separate models; *p*-threshold = 0.05/03 = 0.017) revealed that a difficult infant temperament was positively associated with childhood Negative Affectivity (β = 0.15, *SE* = 0.05, *p* = 0.002), as seen in Figure 1. The association between an easy infant temperament and childhood Negative Affectivity was not significant after Bonferroni correction (β = −0.06, *SE* = 0.03, *p* = 0.031). Neither easy or difficult infant temperaments were significantly associated with childhood Surgency or Effortful Control, all *p*s > 0.017.

### 3.3. Moderating Effect of Parenting Style on the Association between Infant and Childhood Temperament

Two different multivariate model comparisons for each Parenting Scale’s subscale (Laxness, Overreactivity and Verbosity) were performed, to separately test if the model including (a) parenting and its interaction with easy temperament, or (b) parenting and its interaction with difficult temperament, significantly explained more variance than the original model predicting childhood temperamental outcomes. Results show that only the multivariate model including the interaction between an easy infant temperament and Laxness was significantly more predictive than the null model excluding parenting variables (*F*(6,182) = 3.54, *p* = 0.002) (passing Bonferroni correction for six models at a threshold of *p* = 0.008). No significant associations were found for models which tested the interaction between difficult infant temperament and any parenting measure, and easy infant temperament and Verbosity or Overreactivity.

Follow-up tests were conducted only for the winning model (see Table 4 and Table 5), specifically investigating the effect of Laxness on the trajectory between easy infant temperament and childhood temperament (Negative Affectivity, Surgency and Effortful Control). Results showed that the multivariate effect was due to a significant effect on childhood Negative Affectivity, but not on Surgency or Effortful Control. There was a significant main effect of Laxness (β = −1.08, *SE* = 0.28, *p* < 0.001), showing that higher levels of parental Laxness were associated with less childhood Negative Affectivity. There was also a significant main effect of easy infant temperament on Negative Affectivity (β = −0.72, *SE* = 0.17, *p* < 0.001), indicating that an easier infant temperament was related to reduced childhood Negative Affectivity. However, this relationship was moderated by parental Laxness, as indicated by a significant interaction between easy infant temperament and Laxness (β = 0.2, *SE* = 0.05, *p* < 0.001). As shown in Figure 2, the association between easy infant temperament and childhood Negative Affectivity becomes more positive with increasing Laxness, such that children of parents who rated themselves as laxer showed a positive association between an easy infant temperament and childhood Negative Affectivity. Conversely, for children whose parents considered themselves as less lax, a higher easy infant temperament was associated with a lower level of childhood Negative Affectivity. Simple slope analyses revealed that the effect of an easy infant temperament on childhood Negative Affectivity was significantly negative for low parental laxness both at −1.5 SD (β = −0.29, *p* < 0.001) and −1 SD (β = −0.21, *p* < 0.001) from the mean. For high parental laxness, the association between easy infant temperament and childhood Negative Affectivity was non-significantly positive at +1 SD (β = 0.09, *p* = 0.085) and significantly positive at +1.5 SD (β = 0.16, *p* = 0.014) from the mean.

## 4. Discussion

This study aimed to investigate the association between maternal perceptions of infant and childhood temperament and the potential intervening role of parenting styles on these trajectories. Results show that maternal reports of difficult infant temperament are predictive of the childhood temperament dimension of Negative Affectivity at 4–7 years. Furthermore, easy infant temperament interacts with parental laxness in predicting Negative Affectivity in childhood. Results referring to temperamental predictability and the influence of parenting style will be discussed separately.

In line with our hypothesis of an association between infant and childhood temperament, difficult infant temperament showed a significant positive association with childhood Negative Affectivity. This association is in line with the literature on temperamental stability over time in samples of term-born children. For example, Guerin and Gottfried [59] reported that parents maintain a stable perception of their children’s temperament, such that a difficult temperament in infancy predicts negative mood and slow adaptability during childhood. Similar results were discussed by Rothbart et al. [60], who found that laboratory measures of irritability and frustration at 10 months predicted anger, frustration, discomfort, aggression, risk-taking pleasure and guilt–shame at 7 years. This consistency in adverse temperamental qualities suggests that “difficult children” have an early vulnerability for developing and maintaining later adverse behavioural outcomes such as externalising and internalising problems [5,61,62,63,64,65]. Although only a few studies to date have investigated these questions in preterm children, Gracioli and Linhares [66] reported early childhood frustration as one of the main predictors for both externalising and internalising behaviour problems in preterm children. Given the association between difficult temperamental traits in infancy and childhood in our sample, one might speculate that an early difficult temperament contributes to the emergence of the “preterm behavioural phenotype”. Alternatively, it may be an indicator of underlying brain alterations implicated in the emotional development of preterm children [41]. This highlights the importance of considering temperamental traits early in development, as well as the need for further studies exploring the biological substrates of a difficult temperament in preterm children. However, as our study did not have a control group, we cannot exclude the possibility that the effects observed here are not unique to the preterm population. Indeed, some studies have shown that preterm children did not differ from full-term children in terms of difficult temperament, or Negative Affectivity (for a review, [35]). It is therefore possible that difficult temperamental traits follow a similar relatively stable developmental trajectory in both preterm and full-term children. 

Notably, an association between easy infant temperament and childhood temperament traits emerged only when taking levels of parental laxness into account. Contrary to our expectation, this temperament-by-parenting interaction was not in line with our hypothesis, according to which children manifesting high levels of negative emotionality would be more vulnerable to negative environmental influences. Rather, in our sample, those children considered to be easier and more manageable by their mothers appeared more vulnerable to parental influences. Specifically, for very preterm children exposed to lower levels of parental laxness, an easy infant temperament and childhood Negative Affectivity were negatively related, such that easier infants expressed less Negative Affectivity in childhood. Conversely, the association was reversed for children whose mothers self-reported themselves as being laxer, such that an easier infant temperament was increasingly associated with higher childhood Negative Affectivity scores as parental laxness increased. 

Laxness reflects a disciplinary style characterised by inconsistent reprimands and permissiveness [54], and lax parenting has been found to be related to oppositional behaviour and conduct disorders, as well as to internalising difficulties [67,68,69]. Our results suggest that, for easy infants who are raised in a non-dysfunctional disciplinary context (i.e., low parental laxness), the easy temperamental trajectory follows a positive path characterised by low levels of negative emotionality. This is in line with reports that smiling, laughter and rapid approaches to objects in infancy predict 6–7-year-old approach tendencies and a positive effect [60]. Easy temperament is also related to fewer subsequent negative developmental outcomes and to more adaptive social, cognitive and personality development [70,71]. Non-dysfunctional parenting, as expressed by low laxness, may have a stabilising effect on very preterm infants who display an easy temperament early on, maintaining an adaptive temperamental trajectory that fosters emotion regulation and thus protects them from experiencing high levels of Negative Affectivity. In contrast, lax parenting may contribute to variability in Negative Affectivity in preterm children who were considered as easy in infancy. Overall, this inconsistent and permissive parenting style seems to have a beneficial effect on infants regarded as less easy by mothers but a detrimental effect on those rated as easier.

This counterintuitive relationship can be framed within the concept of “goodness of fit” [72], which posits that it is the interaction between the properties of the environment and the child’s characteristics and style of behaving that determines the course of development. Thus, a more structured, less permissive parenting style may be better suited to easy preterm infants who might be easier to discipline. In contrast, less easy preterm infants may benefit emotionally from a moderately challenging environment characterised by a more permissive parenting style. This is in line with previous findings that high levels of positive parenting can exacerbate inhibition in infants with high negative emotionality [28,32,73]. However, the counterintuitive association between suboptimal parenting and lower Negative Affectivity in less easy infants could also reflect a temporary aspect of a more complex and non-linear developmental trajectory in these children, which may ultimately increase socio-emotional risk. Further research with longer follow-up periods will be necessary in order to characterise the long-term trajectories of these children. 

The fact that easy and difficult infant temperaments differ not only in their relationship with childhood temperament but also in their interaction with parenting is of particular interest. Here, it is important to note that maternal perception of easy temperament may be reflective of a wide range of characteristics, from resilience and positive emotionality to withdrawal, passiveness or slower development. Further investigations into the meaning of easy temperament following preterm birth are therefore warranted. However, our results converge with previous findings that highly manageable toddlers are the ones most influenced by maternal control and by non-parental day-care quality [74,75]. In contrast, it has been suggested that a difficult temperament may not constitute vulnerability per se, but rather a factor contributing to non-malleability in children [75], which aligns with our observation of relative stability of this temperamental trait. However, the construct of “difficult” temperament is debated as it may encourage an oversimplified view of temperament [8], although it may nevertheless be a useful construct in a clinical setting as it is easily understandable and recognisable by mothers. Indeed, infant temperament data of this study came from the ePrime questionnaire on infant temperament, a qualitative questionnaire developed for use in a clinical research context. Our findings reporting associations between infant and childhood temperamental traits support the view that even a very simple and rapid questionnaire on maternal perceptions of their infants’ characteristics used as a screening tool in a hospital setting can give a good indication of later temperamental outcomes. 

Findings of this study should be interpreted considering the following limitations. First, due to the lack of a control group, we were unable to compare our sample of very preterm children to their full-term peers, and findings may therefore not be specific to the very preterm population. Furthermore, as this study used only maternal reports of both temperament and parenting style, scores might not accurately reflect the intended characteristics and may be affected by respondent bias. Nevertheless, parental reports of childhood temperament have been shown to be very reliable [76,77]. A further limitation is that parental characteristics were measured only at 4–7 years, and we can therefore only speculate that parenting style was stable between infancy and childhood. It is also possible that infants’ temperamental qualities influenced parenting style over early childhood [17]. Moreover, other parental attributes not captured in our study, such as parental temperament, are also likely to exert an effect on childhood temperament [78], which can be usefully addressed in future research.

## 5. Conclusions

In conclusion, our study adds to previous investigations on longitudinal temperamental trajectories and parental influences on these trajectories in preterm samples. Our results suggest that in very preterm children, specific attention should be paid to infants considered as difficult by their mothers, as difficult traits seem to be stable over childhood, thus constituting a risk factor that potentially adds to their specific cognitive and behavioural vulnerabilities. Attention should also be paid to parenting styles in children considered easy to deal with by their mothers, as they may be more vulnerable to suboptimal parenting styles. Specifically, although in a non-dysfunctional parental environment, easy temperament is predictive of lower levels of negative emotionality over time, the presence of a lax parenting style may reverse this association and contribute to an increase in childhood Negative Affectivity. As parental knowledge and self-efficacy are associated with reduced dysfunctional parenting [79], interventions designed to increase parental awareness and skills may be a useful tool for parents of both difficult and easy preterm infants [80]. Future studies should continue to explore temperamental trajectories in the preterm population, comparing preterm and full-term samples and using a multimethod approach, with the aim of early identification and implementation of parent–child interventions in individuals deemed most vulnerable.

## Figures and Tables

**Figure 1 children-09-00508-f001:**
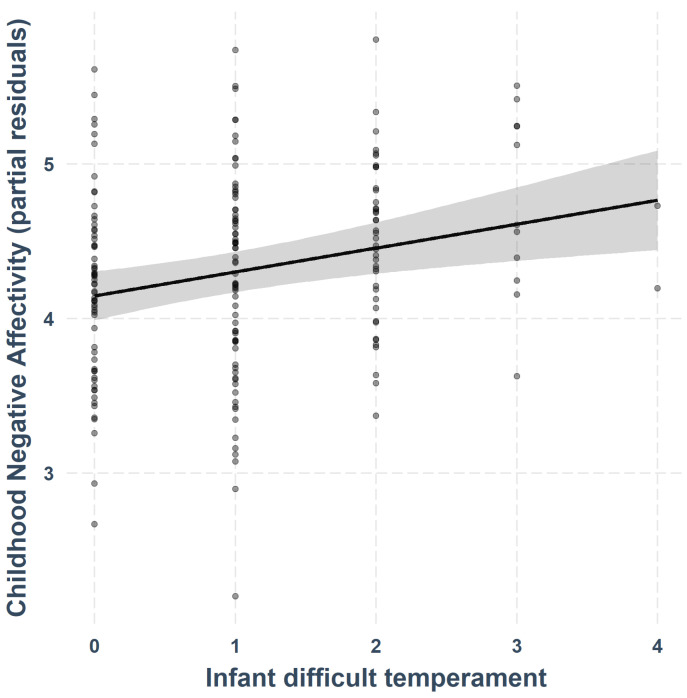
Association between difficult infant temperament and childhood Negative Affectivity (controlling for easy temperament, sex, GA at birth, IMD score and age at follow-up assessment).

**Figure 2 children-09-00508-f002:**
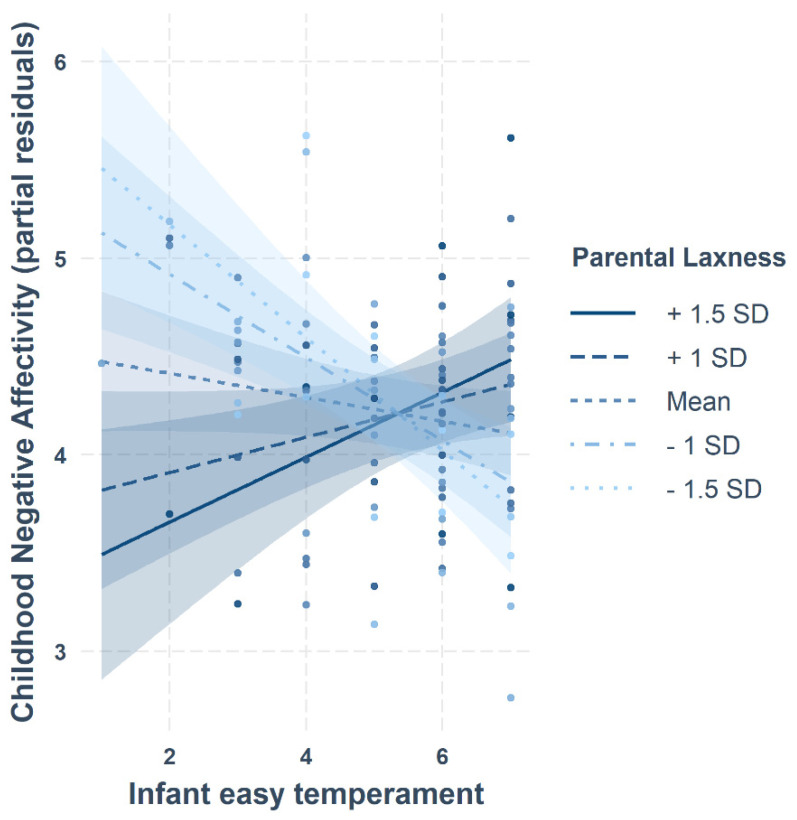
Association between easy infant temperament and childhood Negative Affectivity as a function of parental laxness (controlling for difficult temperament, sex, GA at birth, IMD score and age at follow-up assessment).

**Table 1 children-09-00508-t001:** Study participants’ socio-demographic characteristics.

	Sample with Complete Temperamental Data (*n* = 190)	Sample with Complete Temperamental + Parenting Data (*n* = 101)
Gestational age at birth in weeks, median (range)	30.07 (23.86–32.86)	29.43 (23.86–32.86)
Birth weight in grams, median (range)	1278 (600–2600)	1260 (600–2400)
Female, *n* (%)	88 (46.32%)	41 (40.59%)
Days in hospital, median (range)	45 (3–139)	50 (11–139)
Index of multiple deprivation (IMD) quintile, *n* (%)		
1 (Least Deprived)	46 (24.21%)	27 (26.73%)
2	34 (17.89%)	17 (16.83%)
3	40 (21.05%)	19 (18.81%)
4	49 (25.79%)	29 (28.71%)
5 (Most Deprived)	21 (11.05%)	9 (8.91%)
Corrected age at infant assessment in months, median (range)	11.27 (9–18.4)	11.30 (9.27–18.4)
Corrected age at follow-up assessment in years, median (range)	4.64 (4.18–7.17)	5.18 (4.43–7.17)
Mother’s age at infant’s birth (years), median (range)	33.53 (19.87–53.45)	34.25 (20.08–48.76)
Mother’s age leaving full-time education, at birth, *n* (%)		
16 or less	11 (5.79%)	6 (5.94%)
17–19	26 (13.68%)	9 (8.91%)
19 or more	142 (74.74%)	81 (80.2%)
Still in full-time education	4 (2.11%)	1 (0.99%)
Missing	7 (3.68%)	4 (3.96%)
Maternal ethnicity, *n* (%)		
White	104 (54.74%)	55 (54.46%)
Black or Black British	37 (19.47%)	21 (20.79%)
Asian or Asian British	38 (20%)	19 (18.81%)
Mixed	3 (1.58%)	1 (0.99%)
Other ethnic group	6 (3.16%)	4 (3.96%)
Missing	2 (1.05%)	1 (0.99%)

**Table 2 children-09-00508-t002:** Results of the multivariate regression analysis jointly regressing childhood temperament (Negative Affect, Surgency and Effortful Control) on infant temperament (easy and difficult), controlling for age, sex, gestational age (GA) and index of multiple deprivation (IMD). An uncorrected significance threshold of *p* = 0.05 was used.

	*F*(3,181)	*p*
Easy infant temperament	3.1	0.027
Difficult infant temperament	4.5	0.004
Sex	6.8	<0.001
GA	1.2	0.323
IMD	1.7	0.162
Age	2.4	0.072

**Table 3 children-09-00508-t003:** Results of univariate follow-up analyses separately regressing Negative Affectivity, Surgency or Effortful Control on infant temperament (easy and difficult), controlling for age, sex, gestational age (GA) and index of multiple deprivation (IMD). A Bonferroni-corrected significance threshold of *p* = 0.017 was used.

	Dependent Variable: Negative Affectivity		Dependent Variable: Surgency		Dependent Variable: Effortful Control	
	β	*p*	β	*p*	β	*p*
Easy infant temperament	−0.06	0.031	0.03	0.308	−0.04	0.085
Difficult infant temperament	0.15	0.002	−0.08	0.104	0.02	0.516
Sex	−0.13	0.885	−0.37	<0.001	0.08	0.230
GA	0.03	0.195	0.15	0.465	−0.01	0.565
IMD	0.01	0.030	<0.01	0.851	<0.01	0.272
Age	−0.04	0.490	−0.08	0.179	−0.12	0.013

**Table 4 children-09-00508-t004:** Results of the winning multivariate regression analysis jointly regressing childhood temperament (Negative Affect, Surgency and Effortful Control) on easy and difficult infant temperament, parental laxness and its interaction with easy infant temperament, controlling for age, sex, gestational age (GA) and index of multiple deprivation (IMD). A Bonferroni-corrected significance threshold of *p* = 0.008 was used.

	*F*(3,90)	*p*
Easy infant temperament	4.2	0.008
Difficult infant temperament	3.2	0.027
Parental laxness	1.4	0.250
Parental laxness × easy infant temperament	5.8	0.001
Sex	3.7	0.015
GA	0.7	0.548
IMD	1.5	0.221
Age	1.0	0.387

**Table 5 children-09-00508-t005:** Results of univariate follow-up analyses separately regressing Negative Affectivity, Surgency or Effortful Control on infant temperament, parental laxness and its interaction with easy infant temperament, controlling for age, sex, gestational age (GA) and index of multiple deprivation (IMD). A Bonferroni-corrected significance threshold of *p* = 0.003 was used.

	Dependent Variable: Negative Affectivity		Dependent Variable: Surgency		Dependent Variable: Effortful Control	
	β	*p*	β	*p*	β	*p*
Easy infant temperament	−0.71	<0.001	−0.09	0.645	−0.38	0.009
Difficult infant temperament	0.16	0.015	−0.12	0.097	0.02	0.651
Parental laxness	−1.08	<0.001	−0.10	0.756	−0.58	0.018
Parental laxness × easy infant temperament	0.20	<0.001	0.04	0.460	0.09	0.032
Sex	−0.03	0.789	−0.35	0.008	0.09	0.389
GA	0.03	0.170	−0.01	0.630	−0.10	0.725
IMD	<0.01	0.091	<0.01	0.943	<0.01	0.670
Age	0.01	0.842	−0.07	0.408	−0.09	0.122

## Data Availability

Data and analysis code are available at https://github.com/lucyvanes/preterm-temperament.

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
