# Peer review of "Early Childhood Temperamental Trajectories following Very Preterm Birth and Their Association with Parenting Style"

_children, 2022, doi:10.3390/children9040508_

Round 1

Reviewer 1 Report

Excellent study, easy to read and clear - no suggestions 

Reviewer 2 Report

Thank you for the opportunity to review this manuscript. The current study examined relations between temperament in infancy and temperament in childhood in a sample of very pre-term infants. It also tested the moderating role of parenting style. Temperament by parenting interactions in this population are understudied, thus this research could have an important contribution to the literature. However, I had several questions regarding the analysis plan and the interpretation of the results, including concerns regarding the clarity in the multiple analyses presented, the lack of a comparison/full-term group to contextualize the findings, and whether these analyses actually test stability in temperament. These points and others are detailed below:

Introduction

  1. The sentence beginning on line 55 is difficult to interpret. Are the authors referring to statistical interactions further moderating a link (implying a three-way interaction)? Or rather that parenting moderates the link between infant behavior (e.g., temperament) and later developmental outcomes?
  2. The authors provide evidence for links between different temperament dimensions and parenting styles, but little theoretical discussion is provided regarding reasons why they may be linked. For instance, beginning on line 62, the authors mention that inhibition/shyness is linked to maternal overprotectiveness. What are the processes at play here?
  3. The paragraph beginning on line 67 could use some sharpening. The literature on parenting and temperament is very large, and the authors lead with an example that is counterintuitive to many of the findings in the literature. Both examples are from older studies. Because the goal of the current study is not to elucidate these interactions and make sense of the contradictory findings in the general population (but rather study an extremely preterm population), I would suggest drawing from recent reviews (e.g., Hastings, Rubin, Smith, & Wagner, 2019) or meta-analyses (Slagt, Dubas, Dekovic, & van Aken, 2016) for an overall understanding of these temperament by parenting interactions.
  4. In the discussion of the literature on temperament by parenting interactions for preterm infants, are these findings in comparison to full-term infants? The beginning of this paragraph (line 80) highlights differences between preterm and full-term infants, but once interactions are discussed, either only preterm infants are mentioned or the population is not described. Essentially, is there evidence supporting differences in the temperament by parenting interactions for preterm vs. full-term infants?
  5. I was curious to learn more about the authors’ focus on age 4-7 in the current study, particularly because this is a large age range. It would be helpful for the authors to provide rationale for the importance of studying temperament in both infancy and middle childhood and what changes we might expect to find from a developmental perspective.
  6. Please include hypotheses.

Method

  1. The abstract and introduction define very preterm as before 32 weeks gestation, yet the sample description indicates before 33 weeks. Please clarify.
  2. The authors state that the ePrime questionnaire was used to assess parent perceptions while in the neonatal unit, yet this is an assessment of 12-month temperament. Can the authors describe why this measure was chosen for 12 months, if it is about neonatal perceptions?
  3. Statistical Analysis: I’m not sure I understand what is meant by “The three subscales of the CBQ were jointly included as dependent variables”—what does joint inclusion mean here? I am also unclear about why these preliminary regressions were run before the follow-up regression models. Why not just run multiple regressions that include the infant temperament types, the covariates, and the interactions predicting each CBQ subscale and then correct for multiple comparisons? Alternatively, one could test a hierarchical regression, with the different levels of variables entered in different steps (covariates, followed by 12-month temperament, followed by interactions). Relatedly, the descriptions of the different models are somewhat difficult to follow. This would be alleviated by including model tables in the results. If an issue of space, I think these tables would be more useful than the one scatterplot.

Results

  1. Please include full t-test statistics for the difference between groups regarding child age (line 215).
  2. Please report internal consistency (e.g., Cronbach’s alpha) for questionnaire measures.
  3. Throughout, the first timepoint is labeled as a 12-month timepoint, yet in Table 1, the average age is 11 months and the range is quite large (9-18 months). Perhaps it is more appropriate to call this the infant timepoint rather than a 12-month timepoint.
  4. It does not appear that any effects on surgency or effortful control were reported, significant or non-significant. It would be good to include these relations.
  5. Framing parenting style as an “influence” on the association between infant and childhood temperament (line 234) implies a mediational role. Because an interaction is tested, I would consider rewording this as something like, “The moderating role of parenting style…”
  6. Were simple slopes for both high and low laxness significant? They are interpreted as such, but I do not see significance tests.
  7. I am not sure what to make out of the main effect finding of easy temperament on negative affect in the interaction model (line 252), when earlier, the association between easy temperament and negative affect was not significant after correction (line 229). Again, if analyses are simplified (see previous point), this may be easier to interpret.

Discussion

  1. Because the same temperament dimensions were not tested over time, it is difficult to say that negative affect is “stably predictable.” For instance, difficult temperament, according to this paper, could be described as both “agitated” or “unresponsive and inactive.” Negative affect, in part, is heightened reactivity. It would appear that there is too much measurement variance to say whether a temperament type is truly stable.
  2. Further, I am finding it difficult to contextualize these findings without a comparison group (i.e., full-term infants). Some of these patterns we would expect to find in the general population, so I am not sure whether these findings are unique to preterm infants. For example, I am not sure how early difficult temperament contributes to a “preterm behavioral phenotype” when the literature has established rank order stability of temperament in the general population. The authors note this as a limitation, but it still stands out as a major concern for interpretation.
  3. What is the evidence for stability in easy temperament (line 311-313)? Results suggest that easier temperament is associated with less negative affect in childhood, but I think it may be overstating to say that lower negative affect in childhood evidences stability in easy temperament. This is especially true when easy and negative temperament are not correlated (as is the case here).

Reviewer 3 Report

This research and the results are very interesting and well presented. It is an important study for the important field neonatology and especially the outcome of very preterm infants.

Reviewer 4 Report

Comments for Authors.

I would like to congratulate your work as it pertains to an important issue of temperament/personality development in the first years in preterm children.Your paper has many merits, however my task is to help to improve its quality, so below I list them:

1.Introduction section

I would like Authors to outline in a more convincing way why they decided to conduct this study, that is why they assume that development of temperament of preterm infants may have a different trajectory than development of temperament of term-born children? What specific factors any what kind of mechanisms may occur during prenatal phase of development which may influence temperament development of a child. These factors are oftern a reason for preterm birth, and, as a result, it may cause differences between further temperament development of preterm children and children born at right time. For example, preterm children manifest higher activity level, compared to their term-born peers. How Authors could explain that? Also differences in self-regulation? Generally, in Introduction there should be more information concerning complex underpinnings of temperament development (e.g. role of prenatal stress, toxines, parents’s personality etc.). Also, there should be more information pertaining to the influence of stress on the developing brain and regions responsible for self-regulation which are important issues in this particular topic.

My another suggestions pertains to the fact that Authors did not fully present the development of temperament in childhood in the context of other psychological traits indluded in constructs such as executive function. It is extremely important issue, as some researchers use terms „effortful control” and „executive function/s” interchangeably. Moreover, Authors should provide more information concerning temperaments of parents, which might infere with child’s temperament in different ways (during prenatal and postnatal period).

Summing up, Introduction should be improved by adding more detailed information concerning the specificity of temperament (its different definitions, partial similarities with other psychological constructs; complex underpinnings of its development).

I would also like Authors to provide more theoretical baselines for the questionnaires used in this study (as there are many conceptions of temperament). Why did Authors decide to use these particular tools?

2. Methods

How results obtained in the questionnaires were interpretted? For example, the higher score in Effortful Control means that …? Therefore, I would suggest to provide possible range scores and the way how we interpret intervals.

7-point scale in Parenting Scale – please indicate what 1,2,3,… 7 mean?

Round 2

Reviewer 2 Report

Overall, the authors were responsive to my comments. The resulting manuscript is much improved. However, I have several follow-up points that I would appreciate the authors addressing.

  1. There are a few instances where “12 months,” referring to the infancy timepoint, is left in the manuscript (e.g., line 162).
  2. In the Introduction on temperament, it would be helpful if the authors defined what constitutes an easy vs. difficult temperament. It would also be helpful to define the main outcomes of interest (particularly surgency and effortful control) in the Introduction.
  3. Hypothesis b needs a bit more refinement (line 167). First, is it that more difficult infants would be more vulnerable than easier infants? Second, while the authors bring up differential susceptibility, this test would include both whether more difficult infants are vulnerable to the effects of both positive and suboptimal parenting. The authors do not include a formal test of differential susceptibility. I think it is fine to leave differential susceptibility theory in the introduction, but I would remove it from the hypothesis as it implies a formal test was done (for a helpful paper on these tests, see Del Guidice, 2007). Alternatively, to avoid confusion, the authors could replace differential susceptibility in the introduction and discuss difficult temperament as a vulnerability factor (rather than a susceptibility factor).
  4. CBQ subscales have Cronbach’s alphas greater than 0.66, and the authors labeled this as good internal validity. Typically between 0.6-0.7 is considered adequate/acceptable and good to be above 0.7.
  5. There is a discrepancy in the t-statistic (line 308)—is it p <. 01 or p <. 001?  
  6. I am still struggling with the inclusion of the multivariate analysis (Table 2). Can the authors provide a citation for the exact approach taken and its utility? I don’t quite understand the sentence, “The three subscales of the CBQ were jointly included as dependent variables” (line 267). Does jointly mean that they were combined into one variable somehow? This seems unlikely because “dependent variables” is plural. Or does the multivariate model only tell us that the predictors are related to any of the CBQ subscales, just not which ones? I would also include in the manuscript the multivariate model that includes parenting and the interactions.
  7. Thank you for adding the simple slope analysis, but it is important that these should be used to guide the framing of the Results/Discussion. Because the simple slope for high laxness was not significant, one cannot say (line 360), “The association between infant easy temperament and childhood Negative Affectivity becomes more positive with increasing Laxness, such that children of parents who rated themselves as more lax showed a positive association between infant easy temperament and childhood Negative Affectivity,” as this association was not significant. This should also be addressed in the Abstract, Discussion (line 437), Conclusions (line 540), as many of the interpretations are regarding suboptimal (i.e., high laxness) parenting. The key finding actually seems to be that infants with easier temperaments have lower levels of negative affectivity in childhood when parents are less lax (more optimal).

Reviewer 4 Report

Authors adressed to my concerns.

Author Response

Thank you for your positive comment and for your time reviewing it.

Round 3

Reviewer 2 Report

The authors have been very responsive to my comments. The manuscript is improved. I have no further comments.